# Microbiome Analysis of Traditional Grain Vinegar Produced under Different Fermentation Conditions in Various Regions in Korea

**DOI:** 10.3390/foods11223573

**Published:** 2022-11-10

**Authors:** Haram Kong, Sun-Hee Kim, Woo-Soo Jeong, So-Young Kim, Soo-Hwan Yeo

**Affiliations:** Fermented and Processed Food Science Division, Department of Agrofood Resources, NIAS, RDA, Wanju 55365, Korea

**Keywords:** grain vinegar, acetic acid bacteria, taste fingerprint, metabolic pathway, microbiome

## Abstract

The fermentation of traditional vinegar is a spontaneous and complex process that involves interactions among various microorganisms. Here, we used a microbiome approach to determine the effects of networks, such as fermentation temperature, location, physicochemical and sensory characteristics, and bacterial profile, within traditional grain vinegar samples collected from various regions of Korea. Acetic acid and lactic acid were identified as the major metabolites of grain vinegar, and sourness and umami were determined as taste fingerprints that could distinguish between vinegar samples. *Acetobacter ghanensis* and *Lactobacillus acetotolerans* were the predominant bacterial species, and the functional composition of the microbiota revealed that the nucleotide biosynthesis pathway was the most enriched. These results reveal that vinegar samples fermented outdoors are more similar to each other than vinegar samples fermented at 30 °C, when comparing the distance matrix for comprehending bacterial networks among samples. This study may help obtain high-quality vinegar through optimized fermentation conditions by suggesting differences in sensory characteristics according to the fermentation environment.

## 1. Introduction

Traditional vinegar is a fermented food produced by various microorganisms under complex conditions [1]. Vinegar is used worldwide as an important cooking ingredient and as a preservative [2,3,4]. Due to its organic acid, polyphenol, and melanoidin content, traditional vinegar has various physiological functions, including blood glucose control, lipid metabolism regulation, weight loss, antimicrobial, antioxidant, and anticancer activities [5]. Vinegar is produced from grains or fruits. A dietary culture centered on grain vinegar has been formed in East Asia, especially Korea, along with the development of grain brewing techniques; in contrast, grape and apple fruit vinegar have been developed in Europe [6].

Traditional vinegar fermentation is a spontaneous process; the microbial community associated with vinegar fermentation is naturally enriched and metabolites accumulate. Therefore, it is important to determine the role of microorganisms in acetic acid fermentation (AAF) to improve product quality. Various studies on the bacterial community for AAF have been performed to investigate the functions of fermented vinegar, monitor AAF, and optimize the vinegar fermentation process [1,3,7,8,9,10,11]. Culture-dependent methods for investigating microbial dynamics in AAF enable us to distinguish live cells from dead cells. The bacterial community in traditional vinegar has previously been revealed using culture-dependent methods. *Acetobacter* and *Komagataeibacter* were the dominant genera [11], and *Gluconacetobacter*, *Tanticharoenia*, and *Ameyamaer* were also detected [9]. As it is difficult to meet the conditions required for the growth of specific microorganisms originating from natural habitats through selective culture, it is not simple to cultivate all microorganisms in an ecosystem under laboratory conditions [12]. Consequently, unavoidable limitations are present in culture-dependent methods [13]. Culture-independent methods have also been developed to analyze the 16S rDNA sequences amplified via polymerase chain reaction (PCR) to identify the microorganisms present in the environment, improving the limitations of culture-dependent methods. PCR-DGGE (Denaturing Gradient Gel Electrophoresis) and Illumina HiSeq 16S rDNA sequencing were introduced as microbiome approaches to reveal the microbial dynamics in traditional vinegar. *Lactobacillus* and *Acetobacter* are the predominant genera in traditional vinegar [1,7,8,10,11]. *Klebsiella*, *Gluconacetobacter*, and *Komagataeibacter* were detected via PCR-DGGE, and lactic acid bacteria (LAB), such as *Weissella* and *Leuconostoc*, were detected via Illumina HiSeq sequencing. However, a drawback for 16S rRNA gene sequencing is that it does not provide information about the functional composition of the sampled communities. Therefore, several methods have been developed to predict microbial community functions from profiles of marker gene (16S rDNA) sequencing alone. Phylogenetic investigation of communities by reconstruction of unobserved states (PICRUSt) analysis was developed for the prediction of functions from 16S marker sequences [14,15]. PICRUSt2 integrates existing open-source tools to predict genomes of environmentally sampled 16S rRNA gene sequences and optimizes genome prediction to improve the accuracy of functional predictions. The algorithm used places sequences into a reference phylogeny rather than relying on predictions limited to reference operational taxonomic unit (OTU), bases predictions on a larger database of reference genomes and gene families, predicts more stringently the pathway abundance, and enables predictions of complex phenotypes and the integration of custom databases [15].

The chemical and organoleptic properties of vinegar are a function of the starting materials and the fermentation method [16]. The quality of traditional vinegar depends on its raw materials and the methods employed in its fermentation and manufacturing [17]. In addition, various flavor compounds originating from the complexity of the microbial community provide a rich taste and aroma to vinegar. Acetic acid bacteria (AAB) oxidize carbohydrate alcohols and sugar alcohols (polyhydric alcohols) to their corresponding organic acids, aldehydes, and ketones during AAF [18]. These metabolites, such as organic acids, amino acids, and other flavor compounds, affect the quality of the final product [19]. Fermented vinegar produced through traditional static culture and maturation has acetic acid as its main component, and the non-volatile organic acids, sugars, amino acids, esters, and various volatile substances produced during alcoholic fermentation and acetic acid fermentation give vinegar a unique flavor [20]. Especially, acetic acid is responsible for the taste flavor and pungent, lifting odor of vinegar. The aroma and flavor of vinegars is influenced by the raw materials used, the compounds formed during the fermentation process, and the fermentation method used [16].

The development of next-generation sequencing (NGS) technology enabled us to monitor the quality of fermented foods based on comprehensive omics information regarding microorganisms, additives, and condiments [21]. Metaomic studies of traditional Korean vinegar samples have been reported, yet. Therefore, we analyzed the physicochemical and sensory characteristics of traditional vinegar samples collected from Korea and suggested a microbiological explanation using microbiome sequencing. Furthermore, we discovered differentially abundant genes between metagenomes in traditional vinegar through comparative microbiome and provided information about the community structure, variety, and biological function associated with these genes. The findings of the present study will potentially contribute to the development of the vinegar industry and provide a basis for the production of quality vinegars, thereby increasing the income of the vinegar market.

## 2. Materials and Methods

### 2.1. Traditional Vinegar Collection

Traditional grain vinegar was collected from four regions of the Republic of Korea. The origins of the seven types of grain vinegar used in this study are: CN_UR, Chungcheongnam-do Province and JBG_BR, JBG1_UR, and JBG2_UR, Jeollabuk-do Province both located in the western region of Republic of Korea; GN_BR, Gyeongsangnam-do Province located in the southern region of the Republic of Korea; GB_FG and GB_UR, Gyeongsangbuk-do Province located in the eastern region of the Republic of Korea. The properties of these seven kinds of vinegar are listed in Table 1.

### 2.2. Physicochemical Property Analysis

Titratable acidity was measured as the amount of acetic acid (%) titrated with 0.1 N sodium hydroxide using 1% phenolphthalein as an indicator. The pH was measured using an Orion 3 Star pH meter (Thermo Fisher Scientific, Beverly, MA, USA) at room temperature (20 °C). The total soluble solid (°Brix) was measured using a refractometer (T400321 model, Atago Co., Tokyo, Japan).

### 2.3. Qualitative Analysis of Organic Acid Content Using High-Performance Liquid Chromatography (HPLC)

HPLC analysis for detecting organic acids in the samples was performed using an HPLC system (LC-20A, Shimadzu Co., Kyoto, Japan) equipped with a TSKgel ODS-100V 5 µm (4.6 × 25 cm, TOSOH Co., Tokyo, Japan). The mobile phase consisted of 8 mM perchloric acid (Sigma-Aldrich), the flow rate of the mobile phase was maintained at 1.0 mL/min, and the temperature of the column oven was maintained at 40 °C. The separated material was treated with a reaction solution consisting of 0.2 mM bromothymol blue (Sigma Chemical Co., St. Louis, MO, USA), 15 mM sodium phosphate dibasic dodecahydrate (Na2HPO4, Sigma Chemical Co.), and 7 mM sodium hydroxide (Sigma Chemical Co.). The quantification of the individual compounds was based on the peak areas at 440 nm. Qualitative analysis was performed in triplicate.

### 2.4. Evaluation of Taste Fingerprint Using an Electronic Tongue (e-Tongue)

The taste fingerprint of the collected grain vinegar was analyzed using an e-tongue (ASTREE II, Alpha MOS, Toulouse, France) [22]. The e-tongue included five liquid cross-selective taste sensors (AHS, sourness; CTS, saltiness; NMS, umami; PKS, sweetness; and ANS, bitterness), two reference sensors (SCS and CPS), and an Ag/AgCl reference electrode. Although these seven sensors do not measure the chemical components of taste, they display their intensities on a scale from 0 to 10 [23]. Prior to sample analysis, the e-tongue was prepared by conditioning, calibration, and diagnostics, according to the manufacturer’s procedures. Sucrose (Sigma Chemical Co.) and tannic acid (Sigma Chemical Co.) were used as standards for sweetness and bitterness, respectively. All samples were diluted 1:100 with purified water and filtered through a 0.45-µm membrane filter to remove the solid content and particulates that could affect sample analysis. To prevent contamination between samples, sensors equipped with an e-tongue were washed with purified water before measuring the next sample. Pretreated samples were placed on the instrument sampler and in contact with the sensors for 120 s to analyze their taste fingerprint. The e-tongue test was performed with five replicates. Similar samples were ranked according to the intensity of their salty, acidic, and umami attributes.

### 2.5. Evaluation of Volatile Pattern Using an Electronic Nose (e-Nose)

The headspace of the vinegar samples was analyzed using the HERACLES II electronic nose (Alpha MOS) [24,25]. The HERACLES II was equipped with two identical gas chromatography columns working in parallel mode: a nonpolar column (MXT-5) and a polar column (MXT-WAX) that produced two chromatograms simultaneously. Two columns of MTX-5 and MTX-1701 were mounted in parallel and analyzed using two flame ionization detectors. The temperature of the sample increased at 1 °C per second from the initial temperature of 40 °C to 80 °C while it passed through the column, and from 80 °C, the temperature increased at 3 °C per second and the sample was analyzed until it reached a temperature of 250 °C. The experiment was repeated three times to analyze the difference in the relative content of the components between the samples. The principal component analysis (PCA) and statistical quality control (SQC) analysis was conducted using the AlphaSoft 14.2 ver. software (Alpha MOS).

### 2.6. Culture-Independent MiSeq Microbiome Sequencing

DNA was extracted using a DNeasy PowerSoil Kit (Qiagen, Hilden, Germany), according to the manufacturer’s instructions and the extracted DNA was quantified using Quant-iTTM PicoGreenTM (Invitrogen, Carlsbad, CA, USA). Sequencing libraries were prepared according to the Illumina 16S Metagenomic Sequencing Library protocols to amplify the V3 and V4 regions (Macrogen Inc., Seoul, Korea). Universal primer pairs with Illumina sequences were used for amplification, as follows: V3-F:5′-TCG TCG GCA GCG TCA GAT GTG TAT AAG AGA CAG CCT ACG GGN GGC WGC AG-3′, V4-R:5′- GTC TCG TGG GCT CGG AGA TGT GTA TAA GAG ACA GGA CTA CHV GGG TAT CTA ATC C-3′. The V3 and V4 regions were amplified using only one primer pair. Sequencing was performed by Macrogen, Inc. (Seoul, Korea) using the MiSeq platform (Illumina, San Diego, USA). Illumina MiSeq raw data were sorted by sample using index sequences, and paired-end FASTQ files were created. Sequencing adapter regions and the 16S rDNA F/R primer sequences were eliminated using Cutadapt software (ver. 3.2) [14]. The DADA2 package (ver. 1.18.0) of R software (ver. 4.0.3) was used to perform error correction in amplicon sequencing. The forward (Read1) and reverse sequences (Read2) of the paired-end reads were cut to 250 bp and 200 bp, respectively, and sequences with expected errors > 2 were excluded. The error model for each batch was organized to remove noise from the samples. The error-corrected paired-end sequence was assembled into one sequence, the Chimera sequence was removed using the consensus method in DADA2, and amplicon sequence variants (ASVs) were obtained. For comparative analysis of the microbiota, BLAST+ (ver. 2.9.0) of NCBI was performed in the reference DB (NCBI 16S Microbial DB), and taxon organisms exhibiting high similarity were assigned. In addition, mafft software (ver. 7.475) was used for multiple alignments of each ASV [15].

Various comparative profiles in the microbial community of traditional kinds of vinegar were analyzed using the quantitative insights into microbial ecology (QIIME) pipeline through the abundance and taxon information of the ASVs. To identify species diversity in the vinegar samples, the Shannon index and inverse Simpson index were calculated, and the alpha diversity and mean diversity of species at different sites was evaluated using the rarefaction curve and the Chao1 value. Beta diversity, the ratio between regional and local species diversity, was determined based on weighted and unweighted UniFrac distances.

### 2.7. Phylogenetic Investigation of Communities via Reconstruction of Unobserved States (PICRUSt2) Analysis

PICRUSt2 is a software for predicting functional abundance based only on marker gene sequences [26]. PICRUSt2 analysis was previously used to predict the microbial community metagenome using the Greengene database based on taxonomic abundance [27] and predict the MetaCyc metabolic pathways of the microbiome within samples [26,28]. ASVs with a nearest sequenced taxon index (NSTI) > 2 were excluded from the PICRUSt2 analysis. The function of the microbiota was visualized using ggplot (ver. 3.3.2), and the similarity of clustering features was visualized based on the Bray-Curtis distance [29].

### 2.8. Statistical Analysis

One-way analysis of variance, followed by Duncan’s multiple range test, was used to evaluate the significance of the differences between averages. Statistical significance was set at *p* < 0.05. PCA was applied with maximum variation rotation to visualize significant differences between all data, including organic acids and taste fingerprints. All statistical analyses were performed using SPSS version 17.0 (SPSS Inc., Chicago, IL, USA). Multivariate statistical analyses were performed using the open-source R software (ver. 4.2.1). Pathway and integrated enzyme enrichment analyses were performed and visualized using the pheatmap package (ver. 1.0.12).

## 3. Results

### 3.1. Physicochemical Properties of Grain Vinegar Samples

The physicochemical properties of the collected vinegar samples are shown in Figure 1. The acidity, pH, and the total soluble solid (°Brix) of all samples ranged from 2.1 to 7.3%, 3.0 to 3.9, and 5.6 to10.6 °Brix, respectively. GN_BR exhibited the highest acidity (7.3%) and total soluble solid (10.6 °Brix) among all samples and a pH higher than 3.0. CN_UR exhibited the lowest acidity (2.1%) and the highest pH (3.9) among the seven types of grain vinegar. JBG2_UR exhibited the lowest total soluble solid (5.6 °Brix).

### 3.2. Quantitative Analysis of Organic Acid Content in Grain Vinegar Samples

Various organic acids, such as acetic acid, ascorbic acid, formic acid, lactic acid, propionic acid, and succinic acid, were detected via HPLC analysis of the seven grain vinegar samples (Table 2). Acetic acid and lactic acid were the major organic acid components in the analyzed samples. The proportion of acetic acid among the analyzed organic acids was 91.45% (33,268.91 mg/100 mL). The highest concentration of acetic acid was found in JBG2_UR (7644.36 mg/100 mL), followed by JBG_BR and JBG1 UR (6971.23 mg/100 mL and 6457.86 mg/100 mL, respectively). CN_UR possessed the highest lactic acid content (447.3 mg/100 mL) but the lowest content of acetic acid (937.47 mg/100 mL) among all samples. Ascorbic acid, formic acid, and propionic acid were detected only in JBG_BR. Succinic acid was detected the most in JBG_BR, followed by JBG1_UR, but not in GN_BR and GB_FG. The total amount of analyzed organic acids was highest in JBG_BR and JBG2_UR and lowest in CN_UR. This result shows that the organic acids comprising grain vinegar varied according to the manufacturing method used and the region in which they were produced.

### 3.3. Evaluation of Taste Fingerprint Using e-Tongue

Statistical analyses of the taste fingerprints of the seven grain vinegar samples were subjected to PCA (Figure 2A). The first and second PCs (PC1 and PC2) accounted for 95.754% and 3.179% of the variance in the data, respectively, and the variations were divided in PC1. Two groups appeared, owing to the PCA analysis for the seven kinds of grain vinegar. JBG1_UR and JBG2_UR were located in the negative part of the PC1 dimension in the PCA factor loading plot, whereas the other five samples were positively correlated with PC1. Through PCA analysis, a discriminative fingerprint was proposed for JBG_BR, JBG1_UR, and JBG2_UR compared with the other four samples.

Furthermore, discriminative fingerprints were observed in the sensory characteristics of JBG_BR, JBG1_UR, and JBG2_UR (Figure 2B,C). In the case of sourness (AHS), JBG_BR, JBG1_UR, and JBG2_UR exhibited an average intensity of 7.3 (Figure 2C). On average, the other samples showed an intensity interval of approximately 2.2 with the upper group. Saltiness (CTS) was distributed at an intensity range between 4.4 to 6.9, with JBG1_UR and JBG2_UR exhibiting a high average intensity of 6.9. GN_BR, GB_FG, and GB_UR exhibited an intensity of 6.2 on average. JBG_BR and CN_UR had an intensity of 4.9 on average. Umami (NMS) was distributed at an intensity between 4.5 and 9.0. Similar to sourness and saltiness, JBG1_UR and JBG2_UR showed a high average intensity of 9.0. JBG_BR exhibited an intensity of 5.6, whereas the other four samples exhibited an intensity of 4.6 on average, showing a significant interval of about 4.4 from the upper group. Compared to umami, a similar tendency was observed for bitterness (ANS) (Figure 2B). JBG1_UR and JBG2_UR exhibited a high intensity of 8.4 on average, whereas the other five samples showed an average intensity of 5.0, showing a huge interval of approximately 5.0 with the upper group. However, the opposite tendency was observed for sweetness (PKS). JBG1_UR and JBG2_UR showed a low intensity of 3.3 on average, whereas CN_UR, GN_BR, GB_FG, and GB_UR exhibited a high intensity of 7.3 on average.

### 3.4. Evaluation of Volatile Pattern Using e-Nose

E-nose evaluation with the PCA and SQC analysis was used to identify volatility patterns in seven grain vinegars. The PCA plot showed that the separation according to PC1 explained 96.669% of the variability, whereas PC2 was 3.266%, giving an overall value of 99.935% variability, which is sufficient to explain the similarity between samples (Figure 3A). Seven grain vinegars were identified in PC1. Based on the center of PC1 (JBG_BR and GB_FG), GN_BR, GB_UR, and JGB2_UR were located in the positive direction, whereas JGB1_UR and CN_UR were located in the negative direction. This pattern was related to the acidity of the samples. Samples located in the positive direction showed high acidity (approximately 7.0%), and samples located in the negative direction exhibited low acidity (approximately 3.9%). In addition, CN_UR was located in the positive direction in PC2, and JBG1_UR was located in the negative direction. The SQC analysis is a very useful tool for classifying samples and finding differences [30]. The SQC analysis was used to classify seven grain vinegars (Figure 3B). The seven grain vinegars were classified into three types: the upper group, (CN_UR and JBG1_UR), middle group (GB_FG and JBG_UR), and subgroup (GN_BR, GB_UR, and JBG2_UR). Based on the PCA and SQC analysis, the volatile patterns of seven grain vinegars were distinguished.

### 3.5. Bacterial Community Investigation in Grain Vinegar through Microbiome Analysis

Microbiome analysis of seven types of grain vinegar was performed to analyze bacterial abundance and diversity. Bacterial abundance and diversity were expressed as diversity indices, including amplicon sequence variants (ASVs), Chao1 richness estimate, Shannon index, and inverse Simpson index (Appendix A). An ASV is a taxonomic unit, the Chao1 richness estimate is the total number of species in a community, and the Shannon index is a population index. Higher Shannon index values indicate the presence of diverse microbial communities [31]. In total, 2773,034 DNA reads were detected in the seven grain vinegar samples, representing 242 ASVs. The ASV and Chao1 richness estimates were highest in GN_BR, whereas the Shannon index was highest in JBG1_UR. Therefore, it was assumed that the bacterial diversity varied between GN_BR and JBG2_UR. Good’s coverage values were >0.9997 among all the samples, indicating that the identified sequences represent the microbiome in the analyzed grain vinegar samples [19].

The sequencing reads for bacteria were classified at the genus and species levels to compare the microbial community diversity of the seven grain vinegar samples (Figure 4). In total, 104 bacterial genera were identified. The predominant groups in the samples were as follows: genus level, *Acetobacter* and *Lactobacillus*; species level, *Acetobacter ghanensis*, *Lactobacillus acetotolerans*, *Lactobacillus johnsonii*, and *Lactobacillus ultunensis*. In particular, *A. ghanensis* and *L. acetotolerans* were dominant in the seven grain vinegar samples.

*Acetobacter* was the most abundant genus in JBG_BR with 23,533 ASVs, followed by 17,869 ASVs in JBG2_UR, 13,739 ASVs in GN_BR, and 13,726 ASVs in GB_UR. In contrast, *Lactobacillus* was the most abundant in GB_UR with 27,023 ASVs, followed by 25,464 ASVs in GN_BR, 21,344 ASVs in JBG2_UR, and 16,398 ASVs in JBG_BR, exhibiting the opposite tendency as in *Acetobacter*. Furthermore, microbiome analysis revealed the presence of *Bacillus* spp. (*B. fungorum*, *B. velezensis*, and *B. aerius*) and *Streptococcus* spp. (*S. periodonticum*, *S. lutetiensis*, *S. lactarius*, and *S. gordonii*) in the grain vinegar samples analyzed.

### 3.6. Predictive Functional Genes of the Microbiome in Grain Vinegar

Hierarchical clustering using the R package was applied to identify groups of samples with similar functional pathways (Figure 5A and Appendix A). The seven grain vinegar samples were segregated into seven groups. The nucleotide biosynthesis pathway had a proportion of 19.4%, representing the highest enrichment. Energy metabolism related to the tricarboxylic acid cycle (TCA) cycle accounted for 13.6%, followed by amino acids, fatty acids, carbohydrates, cell wall synthesis, vitamin synthesis, and second metabolism synthesis pathways, accounting for 11.9%, 11.1%, 6.9%, 6.6%, 5.1%, and 4.6%, respectively.

The PWY-7208 pyrimidine pathway and PWY-7219 and 7221 purine pathways, which are probably related to cell growth, were enriched in the nucleotide synthesis pathway, indicating a rapid accumulation of microorganisms during the acetic acid fermentation process. PENTOSE-P-PWY and NONOXIPENT-PWY are pathways involving the highest number of functional genes in energy metabolism, indicating that microorganisms metabolize rapidly and release a large amount of energy to drive the energy-demanding reaction due to stress conditions during acetic acid fermentation.

The visual distribution of the critical conversion pathways in grain vinegar was evaluated based on 16S rRNA gene sequencing data and metabolic profiles inferred using PICRUSt analysis. Figure 5B shows the major metabolic pathways and the related genes involved in producing organic acids. Alcohol dehydrogenase, which catalyzes the conversion of aldehyde to ethanol, and aldehyde dehydrogenase, which catalyzes the conversion of aldehyde to carboxylic acid, are considered directly involved in the production of organic acids [32]. Acetyl-CoA and pyruvate are the main compounds in the metabolic network of the grain vinegar microbiome [33]. Acetyl-CoA participates in the TCA cycle and the biosynthesis of ethanol and acetate. Several organic acids, including succinate, oxaloacetate, lactate, acetate, and butanoate, are also produced in the TCA cycle. In addition, pyruvate is an important intermediate for the formation of lactate and formate.

The distance matrix of the seven grain vinegar samples obtained via PICRUSt2 analysis is shown as a heatmap (Figure 6). The seven grain vinegar samples were clustered into two groups. Group 1 comprised JBG_BR, CN_UR, and JBG2_UR, whereas group 2 comprised JBG1_UR, GN_BR, GB_FG, and GB_UR. It is interesting to note that the samples in group 1 were fermented in outdoors, whereas those in group 2, excluding JBG1_UR, were fermented at approximately 30 °C (Table 1). Group 1 showed the highest abundance of *Acetobacter* in the culture-independent microbial community analysis among the grain vinegar samples analyzed (Figure 4). In particular, high levels of acetic acid were detected in JBG_BR and JBG2_UR of group 1 in the quantitative analysis of organic acids using HPLC (Table 2). Similarly, taste fingerprinting using the e-tongue revealed that the sourness intensities of JBG_BR and JBG2_UR of group 1 were high. Therefore, it was suggested that fermentation temperature, including outdoors conditions, can affect the physicochemical and sensory characteristics, as well as the microbial community of the final product.

## 4. Discussion

Vinegar can be divided depending on their acidity into low-acidity, 4–5% acetic acid; general acidity, 6–7%; double-strength acidity, 13–14%; and triple-strength acidity vinegar, 18–19% [34]. Most of the grain vinegar samples used in this study had an acidity of more than 5.7%, corresponding to low and general acidity vinegar. Meanwhile, an acidity of 2.1% was measured in CN_UR, with a total organic acid content of 1421.62 mg/100 mL, the lowest level among all samples analyzed in this study (Table 2). According to its manufacturer, CN_UR was bottled at >4.6% acidity. Therefore, it was assumed that the acidity of CN_UR decreased during maturation.

Low pH values were measured in JBG1_UR, but not high levels of acidity. The total organic acid content detected in JBG1_UR was high, at 6981.43 mg/100 mL; a high level of lactic acid, at 419.81 ± 22.44 mg/100 mL, was also detected. This result suggests that other organic acids, aside from acetic acid, contribute to the low pH value of JBG1_UR. In addition, previous metagenome sequencing results indicated that *Lactobacillus* was predominantly detected in the microbial community of JBG1_UR, indicating that *Lactobacillus* affects its lactic acid content. Hence, it is suggested that the presence of *Lactobacillus* during AAF affects the characteristics of the final products.

Meanwhile, some *Bacillus* spp., AAB, and LAB can be considered the functional microbiota responsible for producing various lytic enzymes, substrates for alcoholic fermentation, and flavor compounds [35]. For instance, amylase from *B. amyloliquefaciens* degrades starch into dextrin, maltose, and glucose. In this study, *Bacillus* spp. were detected during the microbiome analysis. GN_BR, wherein *B. fungorum* was detected, exhibited the highest total soluble solid (°Brix) among all the samples analyzed (Table 1), suggesting that *Bacillus* spp. can affect the total soluble solid (°Brix) of the final product through amylase activity.

AAF refers to the oxidation of alcohol to acetic acid by AAB. The microbes that participate in AAF are complex; the AAB participating in AAF typically comprise ten genera in the *Acetobacteraceae* family [36]. The *Acetobacteraceae* family comprises the genera *Acetobacter*, *Acidomonas*, *Ameyamaea*, *Asaia*, *Gluconacetobacter*, *Gluconobacter*, *Granulibacter*, *Komagataeibacter*, *Kozakia*, *Neoasaia*, *Saccharibacter*, *Swaminathania*, and *Tanticharoenia* [37]. Among them, *Acetobacter*, *Gluconobacter*, *Gluconacetobacter*, and *Komagataeibacter* are the major bacteria involved in the oxidation of alcohols to acetic acid [19]. Among AAB, *Acetobacter* is commonly found in vinegar produced using traditional methods [38,39]. It is suitable for vinegar production on an industrial scale because it directly uses cheap ethanol as a substrate and only produces acetic acid as a product [39]. In this study, metagenome sequencing of seven grain vinegar samples revealed the existence of two genera (*Acetobacter* and *Komagataeibacter*) and two species (*A. ghanensis* and *K. xylinus*) in *Acetobacteraceae*, with *Acetobacter* being the dominant species among the AAB.

*Gluconacetobacter* is a major bacterium in AAF [40]. A few *Gluconacetobacter* species can oxidize alcohols into acetic acid at high concentrations of alcohol and acidic conditions and are more resistant to acetic acid than *Acetobacter* species. Meanwhile, recent progress in phylogenomic analysis has consolidated and differentiated closely related *Gluconacetobacter xylinus*, and a novel taxon, *Komagataeibacter* [41]. *Komagataeibacter* was only detected in JBG_BR among all the samples. This supports the hypothesis that JBG_BR possessed a high acetic acid content in the qualitative analysis of organic acids (Table 2).

LAB are the dominant species along with AAB in grain vinegar. LAB generally comprise *Aerococcus*, *Carnobacterium*, *Enterococcus*, *Lactobacillus*, *Lactococcus*, *Leuconostoc*, *Oenococcus*, *Pediococcus*, *Streptococcus*, *Tetragenococcus*, *Vagococcus*, and *Weissella* [42]. *Lactobacillus acetotolerans* was isolated from the fermented broth of rice vinegar and was resistant to high concentrations of acetic acid [43]. *L. acetotolerans* can contribute to the production of organic acids in AAF through metabolic processes such as homofermentative DL-lactic acid production. Lactic acid, produced by LAB, is the major carbon source for *Acetobacter* [44]. Lactic acid is oxidized to acetoin and acetic acid by α-acetolactate, which simultaneously oxidizes ethanol. In addition, LAB (e.g., *Lactobacillus* and *Streptococcus*) can lower the pH through lactic acid fermentation and contribute to the formation of an ideal pH environment (pH 3–5) for the alcohol fermentation of yeast [45]. In this study, the proportion of *Lactobacillus* in JBG1_UR with the total soluble solid of 5.9 °Brix was 61.44%. Hence, the *Lactobacillus* in JBG1_UR might produce a large amount of lactic acid during lactic acid fermentation and induce a low pH. This might affect yeast to spend sugar through alcohol fermentation and lead to low total soluble solid (°Brix) in the final product, i.e., a high proportion of LAB may induce a low total soluble solid (°Brix) in the final product. Notably, CN_UR, wherein a relatively low proportion of *Lactobacillus* was detected (49.52%), exhibited a high total soluble solid (8.0 °Brix).

Six genera of LAB were detected in the grain vinegar samples evaluated in this study: *Aerococcus*, *Lactobacillus*, *Lactococcus*, *Leuconostoc*, *Pediococcus*, and *Streptococcus*. *Lactobacillus* was the dominant bacterium, and the proportion of *Lactobacillus* in GB_UR was 66.32%. Five other LAB, except *Lactobacillus*, were detected at an abundance of <0.19% in each sample, which did not affect the characteristics of the final product. *Lactococcus* may also be derived from alcohol mashes [19]. Nie et al. [1] reported the presence of *Streptococcus* in the early stage of traditional vinegar aging and the presence of *Lactobacillus* in the later stage.

In the evaluation of taste fingerprints using an e-tongue, the sourness (AHS) of JBG_BR, JBG1_UR, and JBG2_UR exhibited a high average intensity of 7.3 (Figure 2). This might have been caused by the high acetic acid content in these samples (Table 2). In particular, many ASVs of *Acetobacter* spp. were detected in JBG_BR and JBG2_UR compared to the other samples. Likewise, the bacterial community could affect the acidity and taste fingerprints of the final products. In the evaluation of sweetness (PKS), CN_UR, GN_BR, GB_FG, and GB_UR exhibited a high intensity compared to the other three samples and showed a high total soluble solid of >7.2 °Brix. Woo et al. [46] reported that the total soluble solid of >7.0 °Brix in the final product indicates incomplete alcohol fermentation. Therefore, the sweetness of CN_UR, GN_BR, GB_FG, and GB_UR may have been caused by incomplete alcohol fermentation.

Unlike the taste fingerprint analysis results in which JBG_BR, JBG1_UR, and JBG2_UR showed a discriminatory tendency in comparison with others, CN_UR and JBG1_UR showed a discriminatory aroma fingerprint in the variation of aroma pattern. It could be hypothesized that an inconsistent trend appeared due to the difference in the analysis methods of e-tongue and e-nose such as the difference in detection methods.

The distance matrix of the PICRUSt2 analysis categorized the grain vinegar samples into two groups according to the fermentation temperature (Figure 6). This suggests that the fermentation location and temperature of traditional vinegar may primarily affect the quality of the final products. These results provide information that can serve as a basis for standardizing the quality of traditional vinegar and the implications for manufacturing better quality vinegar.

## 5. Conclusions

On the basis of a microbiome analysis of traditional Korean vinegars, we established that the properties of vinegars influencing microbial diversity and metabolic activity are dependent on multiple factors, including the operating conditions (location and temperature of fermentation). Distance matrices generated to characterize the bacterial networks among samples indicated that these networks were more similar in outdoor fermented vinegar samples than those of samples produced at 30 °C. The fermented conditions can be attributed to the abundance of acetic acid bacteria and organic acids. In addition, these parameters were found to be positively associated with the sensory characteristics of developed sourness.

## Figures and Tables

**Figure 1 foods-11-03573-f001:**
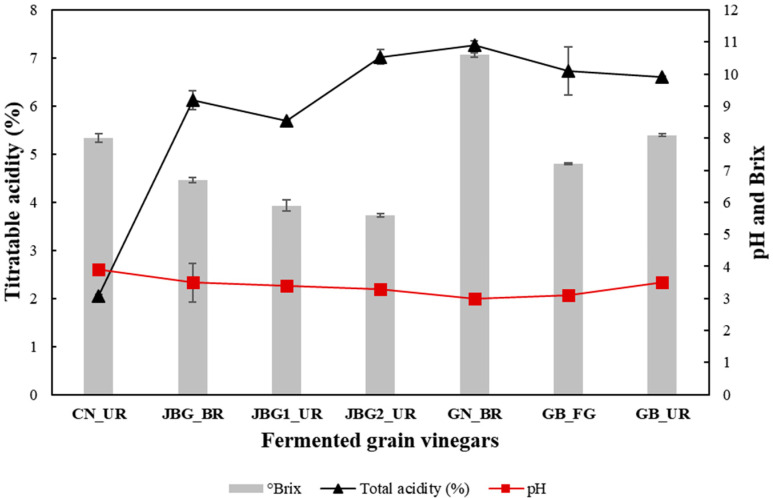
Physicochemical properties of collected grain vinegars. Data are presented as the mean for three independent experiments (*p* < 0.05).

**Figure 2 foods-11-03573-f002:**
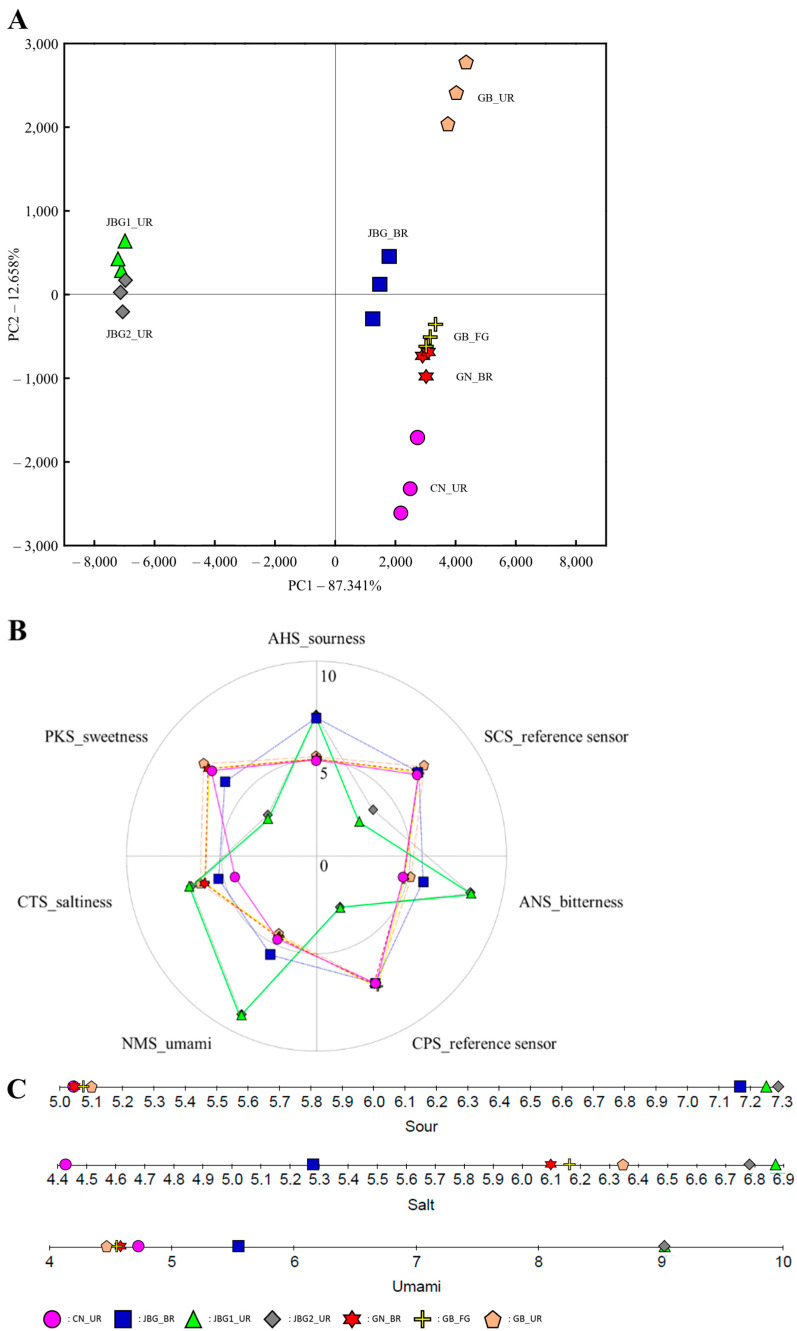
Variation of taste fingerprints in the seven grain vinegar samples analyzed in this study. PCA (**A**), variation in sensory characteristics (**B**), and intensity of sensory characteristics (**C**).

**Figure 3 foods-11-03573-f003:**
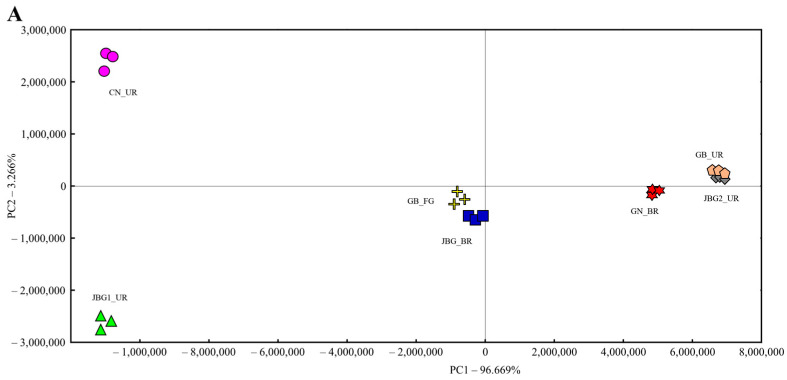
Variation of aroma pattern in the seven grain vinegar samples. PCA (**A**) and SQC (**B**).

**Figure 4 foods-11-03573-f004:**
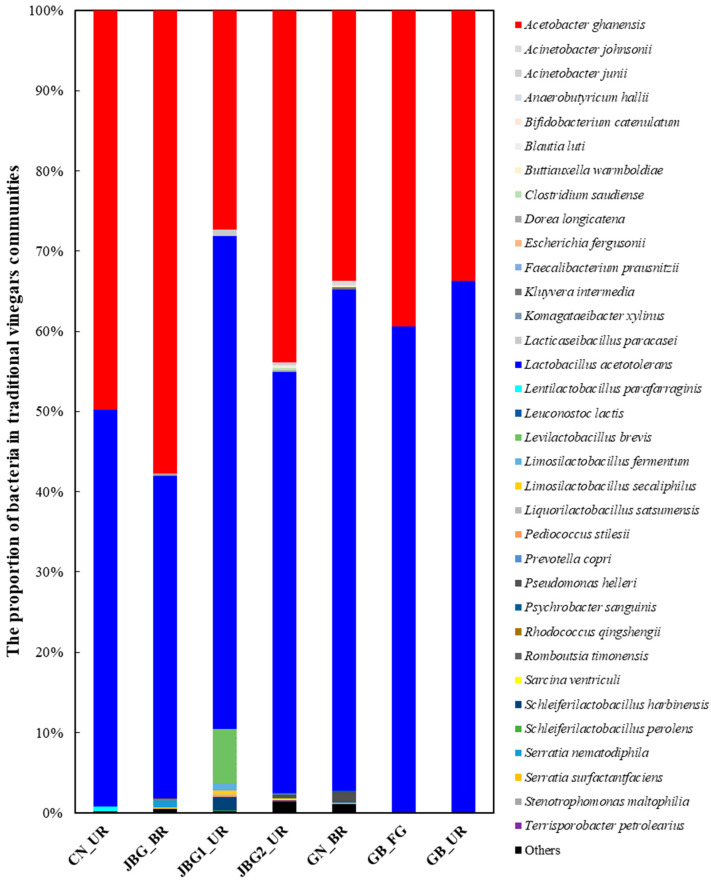
Taxonomic classification at the species level of the bacterial 16S rRNA genes obtained. Species detected below 0.1% were indicated as “others”.

**Figure 5 foods-11-03573-f005:**
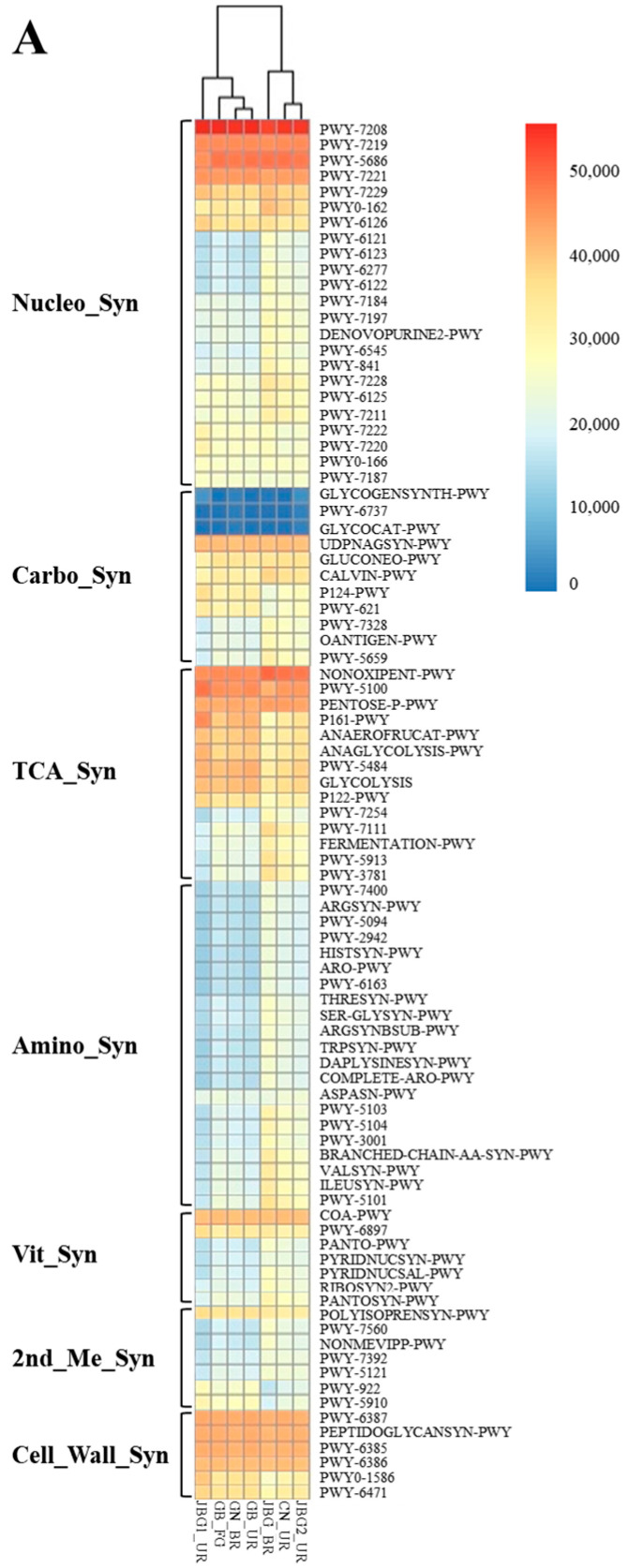
PICRUSt2 analysis of the abundant taxa yielding functional pathways derived from the seven grain vinegar samples analyzed in this study. (**A**) Right: 88 taxa that constituted >1% of the average composition in the enriched group. Pathway names were used to identify the taxa. The phenotypic differences among the clusters are summarized in Appendix A. Each cell in the heatmap is a predicted functional pathway in the grain vinegar samples analyzed. For each vinegar sample listed on the *x*-axis, the relative proportion of the functional pathway (listed vertically along the *y*-axis on the right side of the plot) is represented by a color: blue corresponds to low abundance, and red corresponds to high abundance. See the color key legend at the top left corner. Putative major pathways of organic acid production (e.g., acetate and lactate) in the grain vinegar samples analyzed in this study. (**B**) The putative functions of the microbiota are presented as heatmaps through PICRUSt2 analysis. The pathways were constructed based on the MetaCyc database (P461-PWY, P108-PWY, METH-ACETATE-PWY, PWY-5677, PWY-5100, P124-PWY, ANAEROFRUCAT-PWY, and P122-PWY). The fragments of the heatmap were used to visualize cell values in a color gradient, with red and blue representing high and low confidence, respectively. The order of the samples in the cell was as follows: JBG1_UR, JBG_BR, GB_FG, GN_BR, GB_UR, CN_UR, and JBG2_UR. See the color key legend in the top-right corner.

**Figure 6 foods-11-03573-f006:**
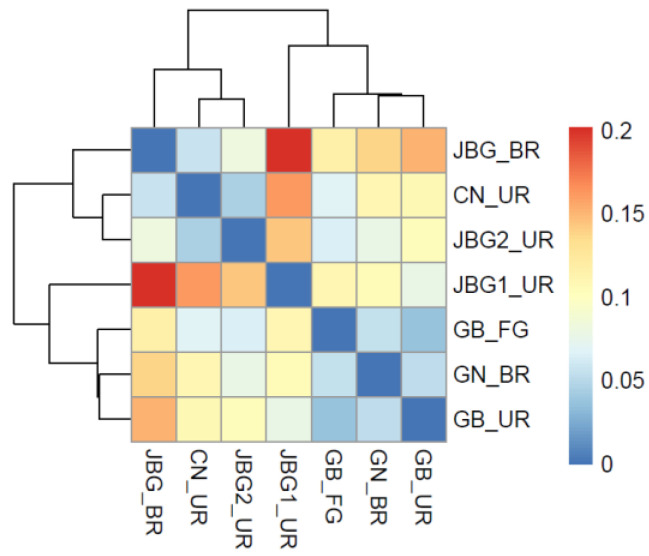
Distance matrix of traditional grain vinegar samples using PICRUSt2 analysis. The distance matrix of the samples is represented by a color: blue corresponds to low abundance, and red corresponds to high abundance. See the color key legend on the right side of the figure. A low abundance means a close relationship between samples.

**Table 1 foods-11-03573-t001:** Information on seven kinds of grain vinegar collected from four regions in Korea.

Region (Province)	Origin	Sample Name	Fermentation Period (Month)	Fermentation Temp. (°C)	Fermentor	Date
Chungcheongnam-do	Brown rice	CN_UR	3	17–25 (Outdoors)	Pottery	Spring 2020
Jeollabuk-do	Black barley	JBG_BR	3	17–25 (Outdoors)	Pottery	Spring 2020
	Brown rice	JBG1_UR	1	14–17	Pottery	Winter 2021
	Brown rice	JBG2_UR	– ^1^	17–25 (Outdoors)	Pottery	Spring 2021
Gyeongsangnam-do	Black rice	GN_BR	–	30	Stainless	Winter 2020
Gyeongsangbuk-do	Five grains ^2^	GB_FG	1	30	Pottery	Winter 2021
	Brown rice	GB_UR	–	32	Pottery	Winter 2021

^1^ Unknown. ^2^ Mixture of brown rice, barley, sorghum, millet, and glutinous millet.

**Table 2 foods-11-03573-t002:** Quantitative analysis of organic acid content in the collected grain vinegar samples (mg/100 mL).

Sample	Acetic Acid	Ascorbic Acid	Formic Acid	Lactic Acid	Propionic Acid	Succinic Acid	Sum
CN_UR	937.47 ± 11.48 ^1,f^	0	0	447.33 ± 6.69 ^a^	0	36.82 ± 0.44 ^c^	1421.62
JBG_BR	6971.23 ± 74.45 ^b^	522.59 ± 28.42	34.25 ± 6.56	23.99 ± 20.91 ^e^	383.75 ± 11.95	224.77 ± 10.93 ^a^	8160.58
JBG1_UR	6457.86 ± 82.59 ^c^	0	0	419.81 ± 22.44 ^b^	0	103.77 ± 20.13 ^b^	6981.43
JBG2_UR	7644.36 ± 61.06 ^a^	0	0	72.22 ± 8.46 ^d^	0	51.12 ± 15.12 ^c^	7767.69
GN_BR	3936.49 ± 68.15 ^d^	0	0	410.98 ± 4.74 ^b^	0	0	4347.47
GB_FG	3622.07 ± 6.45 ^e^	0	0	273.78 ± 6.32 ^c^	0	0	3895.84
GB_UR	3699.44 ± 32.83 ^e^	0	0	73.57 ± 3.73 ^d^	0	33.57 ± 2.26 ^c^	3806.58

^1^ Values are expressed as the mean ± standard deviation (number of replicates = 3). Superscripts in the same column that do not share a common superscript are significantly different at *p* < 0.05, as determined by Duncan’s multiple range test. Different lowercase letters between columns represent significant differences between cultivars (*p* < 0.05).

## Data Availability

Data are contained within the article.

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
