# Peer review of "Microbiome Analysis of Traditional Grain Vinegar Produced under Different Fermentation Conditions in Various Regions in Korea"

_foods, 2022, doi:10.3390/foods11223573_

Round 1
Reviewer 1 Report
1. The manuscript uses bacterial (16S)amplicon sequencing, but the title is "metagenomic analysis", which is misleading because of inaccurate expression.
2. The reads obtained by 16s amplicon sequencing can be classified at the genus levels. However, species-level information obtained by amplicon sequencing is inaccurate, especially for functional prediction. Functional prediction by whole metagenomic sequencing is more accurate than amplicon sequencing.
3. For sensory characteristics, the authors only analyzed taste through the electronic tongue. Why not analyses the odor or flavor?
4. The result “fermentation temperature, including open field conditions, can affect the physicochemical and sensory characteristics, as well as the microbial community of the final product.” is known, but what causes the different sensory properties of vinegars? Which microbes are involved? These are the things the author should analyze in depth.
Author Response
Dear Reviewer,
Please see the attached file for response letter.
Best regards,
Soo-Hwan Yeo

Reviewer 2 Report
The current manuscript reports a metagenomic analysis of traditional grain vinegar produced under different fermentation conditions in various regions in Korea.
In general, this is an important and interesting research, logically structured. The research methods used are described in detail, the results are discussed. I have only one remark, in the Discussion section, the name of the bacteria must be italicized.
Author Response
Response letter to the comments from the reviewer #2
â—‡ Journal: Foods
â—‡ Manuscript ID: foods-1950501
â—‡ Title: Microbiome Analysis of Traditional Grain Vinegar Produced under Different Fermentation Conditions in Various Regions in Korea
Dear Editor,
We would like to express our gratitude to the reviewers who gave their opinions through a fair evaluation regarding this thesis. The reviewers' comments and their responses are attached as follows. The appropriate changes made in the revised manuscript are red.
- Does the introduction provide sufficient background and include all relevant references? : Yes
(No instructions)
- Are all the cited references relevant to the research? : Yes
(No instructions)
- Is the research design appropriate? : Yes
(No instructions)
- Are the methods adequately described? : Yes
(No instructions)
- Are the results clearly presented? : Yes
(No instructions)
- Are the conclusions supported by the results? : Yes
(No instructions)
Comments and Suggestions for Authors
- The current manuscript reports a metagenomic analysis of traditional grain vinegar produced under different fermentation conditions in various regions in Korea.
In general, this is an important and interesting research, logically structured. The research methods used are described in detail, the results are discussed. I have only one remark, in the Discussion section, the name of the bacteria must be italicized.
→ As you said, we found that the italics for the scientific names of microorganisms were removed during upload in some sections, and could have revised them to be italicized.
Furthermore, the manuscript uses bacterial 16S amplicon sequencing, and the title is "metagenomic analysis", which is misleading because of inaccurate expression. Thus, we revised the “metagenomics” to the “microbiome”. We changed that in the title and on the whole manuscript.
The comments made by the member have been corrected. Thank you for your detailed comments to improve the completeness of this manuscript. We believe that these modifications have strengthened the manuscript and hope that the revised manuscript is suitable for publication in Foods.
Sincerely,
Soo-Hwan Yeo
Reviewer 3 Report
Dear Authors,
Please see the attached file for comments and suggestions.
Best regards,

Author Response
Response letter to the comments from the reviewer #3
â—‡ Journal: Foods
â—‡ Manuscript ID: foods-1950501
â—‡ Title: Microbiome Analysis of Traditional Grain Vinegar Produced under Different Fermentation Conditions in Various Regions in Korea
Dear Editor,
We would like to express our gratitude to the reviewers who gave their opinions through a fair evaluation regarding this thesis. The reviewers' comments and their responses are attached as follows. The appropriate changes made in the revised manuscript are red.
- Does the introduction provide sufficient background and include all relevant references? : Can be improved
→ As you said in comment 4, we revised and added an Introduction section of the manuscript (lines 69-70 and 77-83) about “What causes the different sensory properties of vinegars”. It is known that non-volatile organic acids, sugars, amino acids, esters, and various volatile substances produced during alcoholic fermentation and acetic acid fermentation impart the unique flavor of vinegar. Furthermore, the Introduction section was revised to provide an understanding of 16S-based amplicon sequencing and PICRUSt analysis (2p, lines 56-68).
Furthermore, the manuscript uses bacterial 16S amplicon sequencing, and the title is "metagenomic analysis", which is misleading because of inaccurate expression. Thus, we revised the “metagenomics” to the “microbiome”. Changed the title and manuscript.
- Are all the cited references relevant to the research? : Must be improved
→ Please see the response to comment 3. Additionally, references no. 15, 16, and 20 have been added to support additional content. The reference numbers that follow have been revised.
- Is the research design appropriate? : Yes
(No instructions)
- Are the methods adequately described? : Can be improved
→ Please see the response to comment 2.
- Are the results clearly presented? : Yes
(No instructions)
- Are the conclusions supported by the results? : Must be improved
→ Please see the response to comments.
Comments and Suggestions for Authors
- Scientific name of microorganisms must be revised. (Acetobacter, Lactobacillus)
→ As you said, we found that the italics for the scientific names of microorganisms were removed during upload in some sections, and could have revised them to be italicized.
- Materials and methods:
2.1 Line 78: four regions?
→ Thanks to your comments, we were able to revise “three regions” to “four regions” on line 98.
2.2 Line 78-82: Please report more information of sample collection such as season or year period (This factor may affect the microbial diversity).
→ We agree that sample collection may affect microbial diversity. The date of samples was added to table 1 in the manuscript (line 105). Three samples, CN_UR, JBG_BR, and JBG2_UR, fermented outdoors were manufactured in spring. The others, JBG1_UR, GN_BR, GB_FG, and GB_UR, were manufactured in winter. It may support that date affects the quality of the fermented vinegar such as organic acid contents, taste patterns, and microbial diversity.
2.3 More information of the areas where provide traditional vinegar samples is need for more clearly understand the scope of sample collection. Four provinces are in the same region (north/ east/ south/ west) of Republic of Korea or not?
→ As you said, we added the areas that provide traditional vinegars for added clarity (lines 100-103). Chungcheongnam-do and Jeollabuk-do Province are located in western Republic of Korea. Gyeongsangnam-do Province are located in southern of Republic of Korea, and Gyeongsangbuk-do Province are located in eastern of Republic of Korea.
2.4 Table 1: Please report the range of temperature for outdoor fermentation. It is necessary for comparison as the authors report that temperature of fermentation is one of the factors relate to microbial diversity and metabolic activity.
→ For comparison of fermentation temperature, information on outdoors temperature was revised in table 1 of the manuscript (line 105). During fermentation, the outdoor temperature of spring was 17-25 degrees. Other samples were fermented at controlled temperatures. Fermentation temperature might affect microbial diversity and metabolic activity.
2.5 o Brix is a measure of the dissolved solid in a liquid. Although it is commonly used to measure sugar content of an aqueous solution but, in vinegar fermentation contains not only sugar but also organic acids and other soluble solids. Line 89: The sugar content (o Brix) should be replaced with “The total soluble solid ( oBrix) …
→ As you said, in order to provide more accurate information, “the sugar content” have been overall replaced with “The total soluble solid (°Brix)” in the manuscript.
- Reference no.18 was not found in the article.
→ In the process of organizing references, there was a mistake of confusing reference no. 18. Thanks for the point, the correct reference no. 18 was replaced on line 127. Since several references have been added, the reference number 18 have been revised to 22.
The comments made by the member have been corrected. Thank you for your detailed comments to improve the completeness of this manuscript. We believe that these modifications have strengthened the manuscript and hope that the revised manuscript is suitable for publication in Foods.
Sincerely,
Soo-Hwan Yeo
Round 2
Reviewer 1 Report
The study analyzed the microbiome of traditional grain vinegar samples collected from various regions of Korea by 16S amplicons sequencing. However, it is not sufficient to use the electronic tongue only to evaluate the sensory characteristics of products.
Author Response
Response letter to the comments from the reviewer #1-Round2
â—‡ Journal: Foods
â—‡ Manuscript ID: foods-1950501
â—‡ Title: Microbiome Analysis of Traditional Grain Vinegar Produced under Different Fermentation Conditions in Various Regions in Korea
Dear Editor,
We would like to express our gratitude to the reviewer who gave their opinions through a fair evaluation regarding this thesis. The reviewer's comments and their responses are attached as follows. We have provided a revised version of the manuscript marked up using the “Track Changes” function.
- Does the introduction provide sufficient background and include all relevant references? : Yes
- Are all the cited references relevant to the research? : Yes
- Is the research design appropriate? : Yes
- Are the methods adequately described? : Yes
- Are the results clearly presented? : Yes
- Are the conclusions supported by the results? : Yes
(No instructions)
Comments and Suggestions for Authors
- The study analyzed the microbiome of traditional grain vinegar samples collected from various regions of Korea by 16S amplicons sequencing. However, it is not sufficient to use the electronic tongue only to evaluate the sensory characteristics of products.
We agree that it is not sufficient to evaluate the sensory properties of products using only the e-tongue. Therefore, the e-nose analysis results included in the previous response letter was added to the manuscript (Materials and Methods section on pg. 4 (lines 142–154), Results section in pg. 8 (line 270–291), and Discussion section in pg. 14 (line 461–465). As a result of the e-nose analysis, the volatile pattern of the seven grain vinegars was divided into three types, which was attributed to the difference in acidity of the samples. As some references were added, the reference numbers after no. 13 was revised.
Thank you for your detailed comments to improve the quality of this manuscript. The comments made by the reviewer have been addressed. We believe that the modifications made have strengthened the manuscript and hope that the revised manuscript is suitable for publication in Foods.
Sincerely,
Soo-Hwan Yeo